# Segment Any Cancer in CT scans through equipping SAM with cross-slice interaction and indicator prompt

Xian Lin[1][0000−0001−8291−4823]*, Zhehao Wang[1][0009−0006−5249−4171]*, Caozhi Shang[1][0009−0001−9491−895X], Junjie Shi[1][0009−0007−1519−0811], Zengqiang Yan[1][0000−0002−2039−3863], and Li Yu[1][0000−0002−5060−2558]

School of Electronic Information and Communications, Huazhong University of Science and Technology, Wuhan 430074, China
z_yan@hust.edu.cn

**Abstract.** CT scanning has become a commonly used imaging method in cancer diagnosis, displaying the anatomical structure of the human body in detail. However, different types of cancer have varying manifestations in CT imaging, posing great challenges for pan-cancer segmentation in CT scans. Recently, SAM has gradually become a landmark in medical image segmentation due to its powerful generality and generalization, providing a new paradigm for universal segmentation. However, current SAM-based segmentation approaches have weak detail perception ability, heavy dependence on manual prompts, and lack of 3D feature interaction. To address these, we have developed a universal cancer segmentation model for CT scans based on the extraordinary segmentation paradigm of SAM. Specifically, a 3D CNN-based U-shaped image encoder and a cross-branch interaction module are developed to increase the detail feature capture and spatial feature interaction of SAM. Besides, a cancer indicator prompt encoder is introduced to remove the dependence of SAM-based approaches on manual prompts. To fully utilize the advantages of SAM-based universal segmentation models and UNet-based specific segmentation models, we have comprehensively considered the prediction results of both, further reducing false positives and omissions in pan-cancer segmentation. In addition, to fully utilize partially annotated data for specific cancers, we use a combination of pseudo labels and partial labels to generate fully annotated data, effectively avoiding data conflict issues. Our method achieved an average score of 30% and 22% for the lesion DSC and NSD on the validation set and the average running time and area under GPU memory-time curve are 18s and 38960MB, respectively.

**Keywords:** Pan-cancer segmentation · SAM · Auto prompt · Foundation model · Cross-slice interaction.

---

* Xian Lin and Zhehao Wang should be considered joint first authors.

## 1   Introduction

Cancer, a disease caused by the loss of normal regulation and excessive prolifer-ation of human cells, poses a serious threat to human health and is one of the leading causes of death worldwide [8,23]. Common cancers include breast can-cer, lung cancer, colon cancer, rectal cancer, prostate cancer, skin cancer, and stomach cancer, covering the whole body [34]. Early diagnosis and screening, as well as timely treatment, can reduce the mortality rate of cancer [33]. Computer tomography (CT) can provide important information on human tissue structure and is widely used in cancer diagnosis and treatment. In clinical practice, radiol-ogists and clinical doctors manually identify and measure abnormal areas based on CT images [3]. However, this manual detection method is time-consuming, labor-intensive, subjective, and highly dependent on experts. Consequently, de-veloping an end-to-end cancer automatic detection algorithm using deep learning has extremely high clinical value [4].

Compared with conventional organ segmentation tasks, cancer segmentation mainly faces the following challenges [22]: (1) complex and variable shapes with weak regularity; (2) Some cancers have low contrast and blurred boundaries; (3) The location has diversity and may exist in multiple locations simultaneously. Compared with specialized segmentation tasks, the general segmentation task of whole-body pan-cancer segmentation has the following challenges [21]: (1) difficulty in collecting high-quality datasets. The training data for pan-cancer segmentation usually comes from different sources, with different purposes and annotated cancers. Most data sources have only partially annotated pan-cancer; (2) The differences in pan-cancer characteristics are prominent. The visual ap-pearance of different types of cancer varies greatly, and the same cancer also has significant differences among individuals.

To fully utilize various data from different sources, some efforts have been made to develop model training approaches under partial labels. Chen et al. [37] co-train a het-erogeneous 3D network on multiple partially labeled datasets with a task-shared encoder. Huang et al. [36] introduce weight-averaged models for unified multi-organ segmentation on few-organ datasets. Xie et al. [45] propose TransDoDNet, which introduces a dynamic head to enable the network to ac-complish multiple segmentation tasks flexibly and can be trained under partially labeled training data. In addition, to identify cancers of various shapes and ap-pearances in complex backgrounds, plenty of deep learning-based approaches have been proposed for cancer segmentation, demonstrating enormous poten-tial [9,24]. However, these models are tailored for specific cancers, and when applied to other types of cancers, new model parameters need to be trained, which brings great inconvenience to the task of whole-body pan-cancer segmen-tation [7]. The segment any model (SAM), a foundation model for universal seg-mentation, has received considerable praise for its excellent segmentation ability across different objects and powerful zero-shot generalization ability [1]. Based on user manual prompts, including points, bounding boxes, and coarse masks, SAM can segment corresponding objects. Therefore, with simple prompts, SAM can effortlessly adapt to various segmentation tasks [43]. This mode can integrate

multiple individual medical image segmentation tasks into a unified framework, greatly facilitating clinical deployment and providing a new perspective for developing the pan-cancer segmentation model.

Due to the lack of reliable clinical annotations, the performance of SAM in the medical field will rapidly decline [17]. Some foundation models adapt SAM to the field of medical image segmentation by tuning SAM on medical datasets [19,18]. However, these approaches, are prone to disrupting the crucial detail features for identifying small objects and boundaries, making it difficult to segment various cancers with complex shapes, weak boundaries, small sizes, or low contrast . Besides, these SAM-based models require the manual provision of task-related prompts to generate target masks, resulting in semi-automatic pipeline segmentation, which is inconvenient when dealing with pan-cancer segmentation tasks.

In this paper, we propose SAMCancer, a pan-cancer segmentation foundation model that supplements local features to SAM to segment any cancer and introduces a task-indicator prompt encoder for realizing end-to-end automatic segmentation. Specifically, SAMCancer consists of the original SAM, a 3D U-shaped CNN module, a cross-branch interaction module, and a task-indicator prompt encoder. To inherit the powerful feature representation capability of SAM, the structure of SAM has been preserved. To better identify cancers with complex shapes, low contrast, and varying sizes in CT images, we introduced a 3D U-shaped CNN sub-network to capture local features and placed it in parallel with the ViT image encoder of SAM. Then, the cross-branch interaction module is strategically positioned between the ViT-branch and the U-shaped CNN-branch to promote their feature representation ability by exchanging their global semantics and local information with each other. In addition, we extend the SAM-based model to an automatic segmentation model by introducing the task-indicator prompt encoder. Experimental results demonstrate the effectiveness of the proposed SAMCancer.

## 2 Method

### 2.1 Preprocessing

Following SAM-based approaches, the 3D CT scan is converted to 2D slices across the coronal plane to match the inputs of SAM-based models. No resampling method is used in our data preprocessing. As the intensity range of CT is usually (-1024, 2048), directly compressing such a large range to (0, 255) may lose valuable information. Therefore, under the guidance of experts, we choose different density windows for objects during training. Specifically, the intensity ranges of tissue and lung are set as (-200, 300) and (-1300, 300). The intensity range of (-200, 300) is used for inference.

### 2.2 Proposed Method

Network architecture: As depicted in Fig. 1, the proposed SAMCancer consists of the original SAM, a 3D U-shaped CNN image encoder, a cross-branch interaction

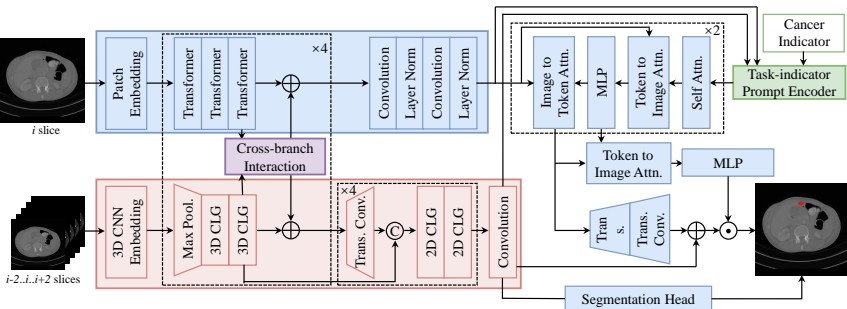

**Fig. 1.** Network architecture. Based on SAM, a 3D U-shaped CNN image encoder and a cross-branch interaction module are proposed to enhance the feature representation ability of the model, and a task-indicator prompt encoder is introduced to realize end-to-end automatic segmentation.

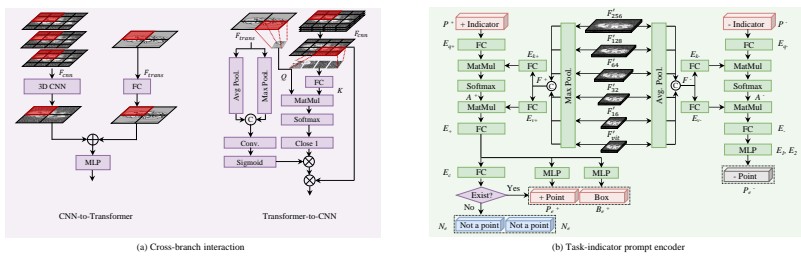

**Fig. 2.** Details of the cross-branch interaction and task-indicator prompt encoder.

module, and a task-indicator prompt encoder. The input of SAMCancer consists of the target slice and its four adjacent slices above and below it. The ViT-branch image encoder is responsible for extracting the feature of the target slice, while the 3D U-shaped CNN image encoder is used to extract abundant detail features of all input slices. By introducing local features of adjacent slices, the model can better cope with cancers with difficult appearances. In addition, we introduce a cross-branch interaction module as a bridge for information exchange between ViT-branch and CNN-branch, as depicted in Fig. 2, to further enhance the feature extraction capability of the model. After encoding image features, a task-indicator prompt encoder (details are presented in Fig. 2) is applied to prompt the mask decoder by the learnable cancer indicators. In addition to the mask decoder, we also adopt a segmentation head to predict the segmentation result based on the output features of the CNN-branch. Finally, the segmentation results of cancer are determined by the combined prediction results of the mask decoder and segmentation head.

Loss function: we use the summation between Dice loss and cross-entropy loss because compound loss functions have proven robust in various medical image segmentation tasks [25].

**Table 1.** Development environments and requirements.

| System | Ubuntu 20.04.6 LTS |
|---|---|
| CPU | Intel(R) Xeon(R) Gold 6143 CPU @ 2.80GHz |
| RAM | 32×8GB; 2666MT/s |
| GPU (number and type) | Two NVIDIA RTX3090 24G |
| CUDA version | 11.1 |
| Programming language | Python 3.8 |
| Deep learning framework | torch 1.8.0, torchvision 0.9.0 |
| Specific dependencies | N/A |
| Code | https://github.com/xianlin7/SAMCT |

Other strategies: We reduce false positives on CT scans from healthy patients by introducing a classifier in the task-indicator prompt encoder. For the inputs of healthy patients, we do not provide positive prompt embeddings for the mask decoder. We use the partial labels to train the model first and obtain the pseudo labels of each scan. Then, we combine the combination of pseudo labels and partial labels to generate the fully annotated data. The whole process is simple, and our focus is more on the model structure. Unlabeled images were not used in our solution. We did not use the pseudo labels generated by the FLARE23 winning algorithm. We did not adopt any additional acceleration strategies for the inference process. However, by using only a small number of adjacent slices, our method can achieve acceptable inference speed and resource consumption.

### 2.3   Post-processing

We directly used the output of the model as the result without any post-processing.

## 3   Experiments

### 3.1   Dataset and evaluation measures

The segmentation targets cover various lesions. The training dataset is curated from more than 50 medical centers under the license permission, including TCIA [6], LiTS [5], MSD [38], KiTS [13,15,14], autoPET [12,11], TotalSegmentator [39], and AbdomenCT-1K [31], FLARE 2023 [30], DeepLesion [42], COVID-19-CT-Seg-Benchmark [28], COVID-19-20 [35], CHOS [20], LNDB [32], and LIDC [2]. The training set includes 4000 abdomen CT scans where 2200 CT scans with partial labels and 1800 CT scans without labels. The validation and testing sets include 100 and 400 CT scans, respectively, which cover various abdominal cancer types, such as liver cancer, kidney cancer, pancreas cancer, colon cancer, gastric cancer, and so on. The lesion annotation process used ITK-SNAP [44], nnU-Net [16], MedSAM [26], and Slicer Plugins [10,27].

The evaluation metrics encompass two accuracy measures—Dice Similarity Coefficient (DSC) and Normalized Surface Dice (NSD)—alongside two efficiency

measures—running time and area under the GPU memory-time curve. These metrics collectively contribute to the ranking computation. Furthermore, the running time and GPU memory consumption are considered within tolerances of 45 seconds and 4 GB, respectively.

**Table 2.** Training protocols.

| | |
|---|---|
| Network initialization | He |
| Batch size | 16 |
| Patch size | 5×256×256 |
| Total epochs | 80 |
| Optimizer | Adam |
| Initial learning rate (lr) | 0.0005 |
| Lr decay schedule | Periodic decay |
| Training time | 120 hours |
| Loss function | Dice loss and cross-entropy loss |
| Number of model parameters | 153.53M[*] |
| Number of flops | 106.83G[*] |
| $CO_2$eq | 26.54 Kg[*] |

**Table 3.** Quantitative evaluation results.

| Methods | Public Validation | | Online Validation | | Testing | |
|---|---|---|---|---|---|---|
| | DSC(%) | NSD(%) | DSC(%) | NSD(%) | DSC(%) | NSD (%) |
| SAMCancer | 27.19 ± 21.20 | 15.07 ± 15.79 | 29.92 | 21.92 | 25.21 | 15.72 |

**Table 4.** Ablation study.

| Methods | Public Validation | | Online Validation | |
|---|---|---|---|---|
| | DSC(%) | NSD(%) | DSC(%) | NSD(%) |
| one slice | 26.72 ± 19.82 | 14.25 ± 13.51 | 27.96 | 18.91 |
| five slices | 27.19 ± 21.20 | 15.07 ± 15.79 | 29.92 | 21.92 |

### 3.2 Implementation details

**Environment settings** The development environments and requirements are presented in Table 1.

**Training protocols** Due to the amount of partial labeled data is large, we did not use unlabeled data. For the partial labels, We divide them into 5 folds for training and obtain pseudo labels for the validation data of each fold, separately. By combining the pseudo labels with partial labels, we obtain fully annotated data. Then, we train the SAMCancer with the fully annotated data. For data augmentation, methods including contrast adjustment, gamma augmentation, random rotation, and scaling are adopted. We take turns using each slice as the target slice and then combine it with its four adjacent slices as a patch after resizing their sizes into 256×256. No special patch sampling strategy was adopted. The best-performing model on the local validation set is selected as the optimal model. More details of the training protocol are presented in Table 2.

## 4  Results and discussion

### 4.1  Quantitative results on validation set

Quantitative results are summarized in Table 3. Our method achieves a mean DSC of 27.19% and a NSD of 15.07% on the FLARE 2024 public validation dataset. On the FLARE 2024 online validation dataset, the proposed approach achieves a mean DSC of 29.92% and a NSD of 21.92%.

**Table 5.** Quantitative evaluation of segmentation efficiency in terms of the running them and GPU memory consumption. Total GPU denotes the area under GPU Memory-Time curve. Evaluation GPU platform: NVIDIA QUADRO RTX5000 (16G).

| Case ID | Image Size | Running Time (s) | Max GPU (MB) | Total GPU (MB) |
|---|---|---|---|---|
| 0001 | (512, 512, 55) | 30.45 | 3563 | 69021 |
| 0051 | (512, 512, 100) | 12.99 | 3563 | 22074 |
| 0017 | (512, 512, 150) | 13.44 | 3563 | 22249 |
| 0019 | (512, 512, 215) | 13.71 | 3563 | 23359 |
| 0099 | (512, 512, 334) | 15.87 | 3563 | 31983 |
| 0063 | (512, 512, 448) | 13.18 | 3563 | 22280 |
| 0048 | (512, 512, 499) | 29.67 | 3563 | 79407 |
| 0029 | (512, 512, 554) | 25.26 | 3563 | 64418 |

Ablation study results are summarized in Table 4. Compared to the single slice input of SAM-based methods, the structure that support multiple slice inputs achieve better performance, indicating the effectiveness of the proposed SAMCancer.

### 4.2  Qualitative results on validation set

Qualitative results are depicted in Fig. 3. By injecting more planar and spatial details into SAM, the proposed method can effectively reduce false positives on the background and improve the accuracy of cancer recognition under low

contrast. However, due to the limitations of the dataset, the proposed method performs poorly on discrete small cancers and rare regional cancers.

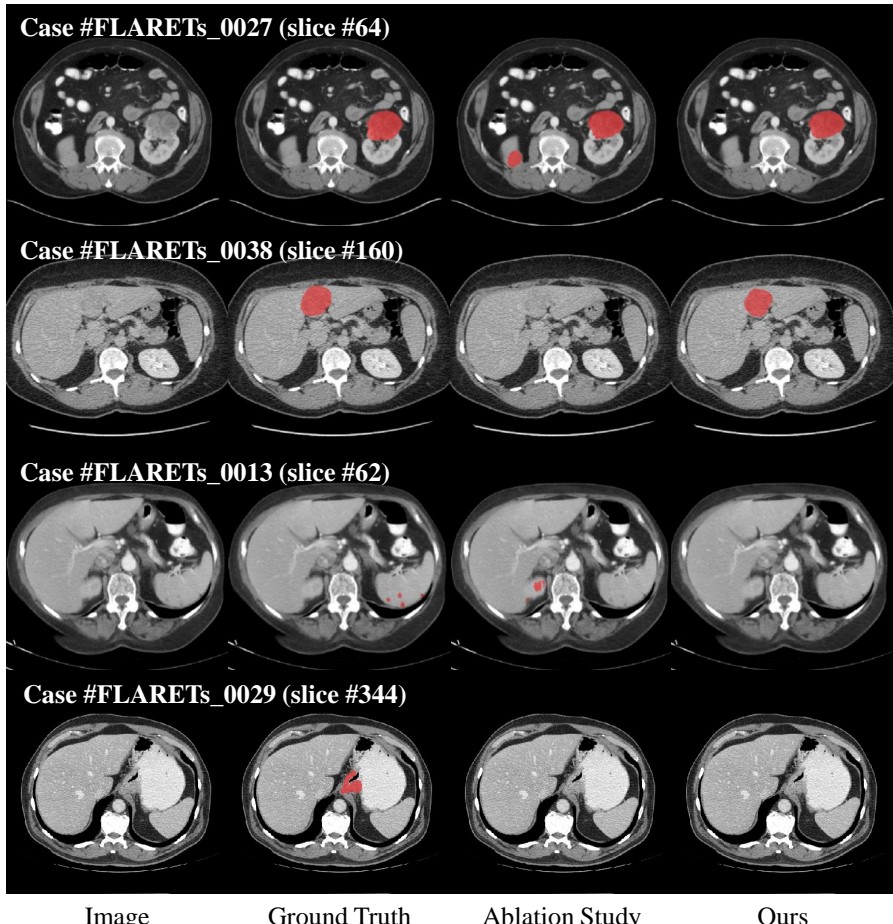

|              |              |              |              |
| :----------: | :----------: | :----------: | :----------: |
| Image        | Ground Truth | Ablation Study | Ours         |

**Fig. 3.** Qualitative results.

### 4.3   Segmentation efficiency results on validation set

The average running time in online validation dataset is 18.05s per case in inference phase, and average used GPU memory is 3563 MB. The area under GPU memory-time curve is 38960. Table 5 lists segmentation efficiency of some typical cases. In addition, the false positive rate of the proposed approach on the healthy CT scans is 0.04 ± 0.05.

### 4.4   Results on final testing set

Quantitative results on final testing set are summarized in Table 3. Our method achieves a mean DSC of 25.21% and a NSD of 15.72%.

### 4.5   Limitation and future work

Our method did not utilize the unlabeled data and did not fully utilize the partially labeled data. In addition, our data preprocessing at the 3D level is not rich. In the future, we will explore how to fully utilize available data for model training under any annotation type. In addition, it is also important for universal medical image segmentation to fully preprocess various 3D data within a unified framework and with the automatic hyperparameter setting.

## 5   Conclusion

In this work, we design a pan-cancer segmentation foundation model based on SAM to segment various cancers. By introducing the 3D U-shaped CNN encoder and the cross-branch interaction module, we can promote the model to recognize various cancers with complex appearances. Besides, introducing the task-indicator prompt encoder makes the SAM-based model an end-to-end automatic pipeline. These designs may be helpful for other universal medical image segmentation tasks.

**Acknowledgements** The authors of this paper declare that the segmentation method they implemented for participation in the FLARE 2024 challenge has not used any pre-trained models nor additional datasets other than those provided by the organizers. The proposed solution is fully automatic without any manual intervention. We thank all data owners for making the CT scans publicly available and CodaLab [41] for hosting the challenge platform.

## Disclosure of Interests

The authors declare no competing interests.

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

**Table 6.** Checklist Table. Please fill out this checklist table in the answer column.

| Requirements | Answer |
| --- | --- |
| A meaningful title | Yes |
| The number of authors ($\leq 6$) | 6 |
| Author affiliations and ORCID | Yes |
| Corresponding author email is presented | Yes |
| Validation scores are presented in the abstract | Yes |
| Introduction includes at least three parts: background, related work, and motivation | Yes |
| A pipeline/network figure is provided | Figures 1&2 |
| Pre-processing | Page 5 |
| Strategies to use the partial label | Pages 5 & 6 |
| Strategies to use the unlabeled images. | Page 5 |
| Strategies to improve model inference | Page 5 |
| Post-processing | Page 5 |
| The dataset and evaluation metric section are presented | Page 5 |
| Environment setting table is provided | Table 1 |
| Training protocol table is provided | Table 2 |
| Ablation study | Page 7 |
| Efficiency evaluation results are provided | Table 5 |
| Visualized segmentation example is provided | Figure 3 |
| Limitation and future work are presented | Yes |
| Reference format is consistent. | Yes |