# OpenReview forum: "Segment Any Cancer in CT scans through equipping SAM with cross-slice interaction and indicator prompt"
_MICCAI.org/2024/Challenge/FLARE — FLARE 2024 withMinorRevisions_

### Official Review · Reviewer_vSa8 · 2025-01-26
**Review of "Segment Any Cancer in CT scans through equipping SAM with cross-slice interaction and indicator prompt“**

**Rating:** 8
**Confidence:** 4

**Review:**

In this paper, a pan-cancer segmentation model based on SAM, SAMCancer, is proposed for cancer segmentation in CT scan images. The method significantly improves the recognition of complex shapes and low-contrast cancers by introducing 3D U-shaped CNNS and cross-branch interaction modules. At the same time, the introduction of task indication prompt encoder realizes end-to-end automatic segmentation and reduces the dependence on manual prompt. Experimental results show that the proposed method performs well on multiple verification sets with high accuracy and efficiency. However, there is still room for improvement in data utilization, such as underutilization of unlabeled data. In addition, the 3D data preprocessing strategy is relatively simple, which can be further optimized in the future.

---

> ### Author Response · Authors · 2025-03-29
> **Response to Reviewer vSa8**
>
> Thank you for your valuable comments. The point-to-point responses are as follows:
> Response 1: Thanks for your advice. We will explore more advanced techniques for leveraging unlabeled data and preprocessing 3D data in the future.

---

### Official Review · Reviewer_iDE7 · 2025-01-27
**Review of  "Segment Any Cancer in CT scans through equipping SAM with cross-slice interaction and indicator prompt"**

**Rating:** 8
**Confidence:** 4

**Review:**

This paper  presents a novel pan-cancer segmentation model, SAMCancer, designed to improve the segmentation of various cancers in CT scans. The method enhances the Segment Anything Model (SAM) by incorporating a 3D U-shaped CNN module, a cross-branch interaction module, and a task-indicator prompt encoder to address challenges such as weak detail perception, heavy dependence on manual prompts, and lack of 3D feature interaction. The study demonstrates the effectiveness of SAMCancer, achieving an average DSC of 27.19% and NSD of 15.07% on the validation set, with an average running time of 18 seconds and GPU memory consumption of 3563 MB.
Here are some minor suggestions:
1. Further Exploration of Data Utilization:
The paper does not fully utilize unlabeled data. Future work could explore more advanced techniques for leveraging unlabeled data, such as semi-supervised learning or self-training, to further improve the model's performance and generalization capabilities.

---

> ### Author Response · Authors · 2025-03-29
> **Response to Reviewer iDE7**
>
> Thank you for your valuable comments. The point-to-point responses are as follows:
> 1. Response 1: Thanks for your advice. We will explore more advanced techniques for leveraging unlabeled data in the future.

---

### Official Review · Reviewer_LsVD · 2025-01-28
**This represents an effective methodology for achieving fully automated tumor segmentation by leveraging SAM, which integrates the global context with the local features of convolutional networks.**

**Rating:** 7
**Confidence:** 4

**Review:**

This study presents a commendable approach to fully automated tumor segmentation, with the paper offering a comprehensive content. Nevertheless, I have several specific inquiries and suggestions:
1、I appreciate your work and would like to inquire if the code corresponding to the methodology described in this paper could be shared and made publicly available, with a clear indication in the manuscript.
2、Regarding the preprocessing step, different thresholding techniques are applied to various organs. Could you elaborate on how the classification of different regions within a CT scan is determined during the inference phase to apply the appropriate preprocessing?
3、Based on my previous experiments, I have observed that 2.5D models may not perform as effectively as 3D models. Have you considered integrating a 3D SAM (Segment Anything Model) to further advance the methodology?

---

> ### Author Response · Authors · 2025-03-29
> **Response to Reviewer LsVD**
>
> Thank you for your valuable comments. The point-to-point responses are as follows:
> 1. Response 1: We have shared the code and provided the link in the manuscript.
> 2. Response 2: Since lesions are usually visible within the soft tissue window, the soft tissue window is mainly used for inference.
> 3. The 3D foundation model is highly computationally complex, and lesions are mainly related to adjacent slices. Therefore, we adopted a 2.5D framework. We will try the 3D framework in the future.

---

### Official Review · Reviewer_VcD2 · 2025-02-17
**Review of "Segment Any Cancer in CT scans through equipping SAM with cross-slice interaction and indicator prompt"**

**Rating:** 8
**Confidence:** 4

**Review:**

This study presents a commendable approach to fully automated tumor segmentation, with the paper offering a comprehensive content. The method enhances the Segment Anything Model (SAM) by incorporating a 3D U-shaped CNN module, a cross-branch interaction module, and a task-indicator prompt encoder, demonstrating the effectiveness of SAMCancer on validation sets.
Nevertheless, I have several specific inquiries and suggestions:
1. 3D Model Consideration:Based on my previous experiments, 2.5D models may not perform as effectively as 3D models. Have you considered integrating a 3D SAM (Segment Anything Model) to further advance the methodology? This might enhance the model's performance in handling 3D data.
2. Further Data Utilization: The paper does not fully utilize unlabeled data. Future work could explore more advanced techniques for leveraging unlabeled data, such as semi-supervised learning or self-training, to further improve the model's performance and generalization capabilities.

---

> ### Author Response · Authors · 2025-03-29
> **Response to Reviewer VcD2**
>
> Thank you for your valuable comments. The point-to-point responses are as follows:
> Response 1: The 3D foundation model is highly computationally complex, and lesions are mainly related to adjacent slices. Therefore, we adopted a 2.5D framework. We will try the 3D framework in the future.
> Response 2: Thanks for your advice. We will explore more advanced techniques for leveraging unlabeled data in the future.

---

### Decision · Program_Chairs · 2025-03-20

**Decision:**

Accept

**Comment:**

Please carefully address the reviewers' comments in the revision.

---

> ### Author Response · Authors · 2025-03-29
> **Response to Program Chairs.**
>
> Thank you for your valuable comment.

---

> > ### Comment · Program_Chairs · 2025-03-31
> >
> > Sec. 4.4 is still not finished